# First Genomic Evidence of a Henipa-like Virus in Brazil

**DOI:** 10.3390/v14102167

**Published:** 2022-09-30

**Authors:** Leonardo H. Almeida Hernández, Thito Y. Bezerra da Paz, Sandro Patroca da Silva, Fábio S. da Silva, Bruno C. Veloso de Barros, Bruno T. Diniz Nunes, Lívia M. Neves Casseb, Daniele B. Almeida Medeiros, Pedro F. da Costa Vasconcelos, Ana C. Ribeiro Cruz

**Affiliations:** 1Parasite Biology in the Amazon Region Graduate Program, Pará State University, Belém 66087-670, Brazil; 2Department of Arbovirology and Hemorrhagic Fevers, Evandro Chagas Institute, Science, Technology, Inovation and Health Strategic Input Secretariat, Ministry of Health, Ananindeua 67030-000, Brazil

**Keywords:** metagenomics, henipavirus, marsupialia, opossums

## Abstract

The viral genus *Henipavirus* includes two highly virulent zoonotic viruses of serious public health concern. *Hendra henipavirus* and *Nipah henipavirus* outbreaks are restricted to Australia and Southeast Asia, respectively. The Henipavirus genus comprises mostly bat-borne viruses, but exceptions have already been described as novel viruses with rodents and shrews as reservoir animals. In the Americas, scarce evidence supports the circulation of these viruses. In this communication, we report a novel henipa-like virus from opossums (*Marmosa demerarae*) from a forest fragment area in the Peixe-Boi municipality, Brazil, after which the virus was named the Peixe-Boi virus (PBV). The application of next-generation sequencing and metagenomic approach led us to discover the original evidence of a henipa-like virus genome in Brazil and South America and the original description of a henipa-like virus in marsupial species. These findings emphasize the importance of further studies to characterize PBV and clarify its ecology, impact on public health, and its relationship with didelphid marsupials and henipaviruses.

## 1. Introduction

Environmental changes associated with socioeconomic factors contribute to reducing barriers between wildlife and human populations. Deforestation causes the destruction of natural habitats and changes in ecosystems dynamics, creating an imbalance in enzootic cycles already well-defined in nature, exposing humans and other animals, including domestic and livestock, to emerging pathogens [1].

Emerging viruses have represented severe public health issues in recent decades, with most of them originated in wild animals such as bats, rodents, non-human primates, and mosquitoes [2,3]. Arboviruses (Dengue, Zika, Chikungunya and Crimean-Congo hemorrhagic fever viruses), coronaviruses (SARS-CoV, MERS-CoV and SARS-CoV-2), filoviruses (Ebola and Marburg viruses), arenaviruses (Sabiá and Lassa fever viruses), and paramyxoviruses (*Hendra henipavirus*—HeV, and *Nipah henipavirus*—NiV) all emerged since the 1990s [3,4].

Paramyxoviruses have a wide host range among vertebrates and some of them already emerged in humans: measles virus, human mumps virus, human parainfluenza viruses, HeV, and NiV. In addition, novel mammalian paramyxoviruses have been frequently identified, some of which may be pathogenic to humans, and under propitious conditions could transpose interspecies barriers and spillover, eventually causing outbreaks [5].

*Paramyxoviridae* is a viral family currently comprising four subfamilies, 17 genera, and 86 species [6]. It is a large group of RNA viruses with non-segmented, negative-sense genomes ranging in length about 14,296 to 20,148 nt. The genomes encode at least six proteins: a nucleocapsid protein (N), a matrix protein (M), a fusion protein (F), a receptor-binding protein (RBP), and a phosphoprotein (P) that associates to a polymerase protein (L) to compose the RNA-dependent RNA polymerase complex [7,8].

*Henipavirus* is a *Paramyxoviridae* genus that comprises some emerging viruses of serious public health concern. HeV and NiV, the main representatives of the viral genus, are highly virulent zoonotic viruses that cause neurological and respiratory diseases and have been reported to be responsible for outbreaks in humans since being first described in Australia and Malaysia in 1995 and 1998, respectively [5,8].

The zoonotic transmission of HeV occurs through contact with infected body fluids from horses, which act as amplifying hosts [9]. On the other hand, transmission of NiV is primarily related to contact with the saliva or urine of infected bats, consumption of meat derived from infected animals and from person-to-person via the respiratory route during outbreaks, which justifies the increasing concern of the emergence of not only NiV but also other henipaviruses [8].

Henipavirus outbreaks are currently restricted to Southeast Asia and Australia. For this reason, studies involving viruses of the genus outside of these geographic regions are less frequent. There are also important serological surveys and phylogenetic studies in both humans and animals conducted in Africa that indicate henipaviruses are circulating on the continent. Moreover, a complete genome of a henipa-like virus was recently classified as a novel henipavirus species, *Ghanaian bat henipavirus* (GhV), with a still unknown zoonotic potential [10,11].

Almost all recognized species in the genus are bat-borne viruses. Bats of the *Pteropus* genus are the primary hosts on Asia and Australia, with *Eidolon helvum*, the fruit bat from which GhV genome was first detected, possibly having an important role in Africa [5,10,12]. However, the *Mojiang henipavirus* (MojV), suspected to have caused human disease in China, was considered a possible exception due to its association with rodents of the *Rattus flavipectus* species [13]. Recently, a novel rodent-borne henipavirus sequence, closely related to MojV, from a *Apodemus agrarius* host became available at the NCBI in September 2021 [14].

Furthermore, novel henipaviruses were described or had sequences submitted to GenBank database that described being found in shrews, mostly of the genus *Crocidura* from China, South Korea, Belgium, and Guinea [14,15,16]. Moreover, a presumably shrew-borne novel henipavirus, phylogenetically related to MojV, referred as Langya henipavirus, was associated with a febrile illness in patients from China, mostly farm workers. The limited epidemiological data analysis indicated that human-to-human transmission was unlikely, and the human cases probably emerged from multiple spillover events [17]. These new data on the diversity of henipavirus in shrews reinforced the possibility that other mammals may harbor henipaviruses instead of only bats and rodents and highlight their potential of emerging as novel human pathogens despite the taxonomic classification of their reservoir animals.

In the Americas, scarce henipavirus surveillance studies have been conducted. Moreover, of a single piece of genomic evidence from Costa Rica [10], two serological surveys in bats from Brazil [18] and Trinidad and Tobago [19] indicate the possibility of circulation of henipa-like viruses on the continent. Here we report the first genomic finding of a henipa-like virus in Brazil and South America, a novel virus called the Peixe-Boi virus (PBV), which corroborates the circulation of henipa-like viruses far from Africa, Asia, and Australia. It is also the first description of henipa-like viruses in marsupials worldwide, indicating the importance of including these animals in surveillance studies.

## 2. Materials and Methods

### 2.1. Samples Collection

A metagenomic study for surveillance of RNA viruses was performed in northeast of Pará State in small wild mammals, with capture authorized by the Ethics Commission on Animal Use of the Evandro Chagas Institute under protocols no. 21/2014 and 40/2019.

In September 2015, 11 small wild mammals (nine opossums and two rodents) were captured in a fragmented forest area adjacent to the rural community of Ananin village in the municipality of Peixe-Boi, Pará State, Brazil (Figure 1C). This region was chosen because of the high deforestation rate, one of the highest in the eastern Amazon (80.24% of its territory as of 2021). Animals were captured using Tomahawk and Sherman traps, which were disposed 10 m distant from each other in three distinct locations: trails inside the forest (A1 to A3), the forest edge (B1 to B3), and in the peripheral area of the community (C1 to C3). They were anesthetized by Zoletil^®^ 50, via the intramuscular route, followed by cervical dislocation euthanization and tissue harvesting (spleen, lymph nodes, heart, and lungs). Samples were stored in liquid nitrogen prior to transport to the Department of Arbovirology and Hemorrhagic Fevers of Evandro Chagas Institute (Ananindeua, Brazil), where they were stored at −80 °C. Initially, metagenomic analysis was conducted for the pooled tissues samples from three opossums (MA7158, MA7165 and MA7168) captured at site A. They were molecularly identified as *Marmosa demerarae* species based on mitochondrial DNA analysis (Figure 1A,B, Appendix A).

### 2.2. RNA Extraction and cDNA Synthesis

Tissue samples were pooled since they were from the same species. A total of 5 mg fragments of the pooled tissues was homogenized in a tube filled with 600 µL of 1-Thioglycerol/Homogenization Solution and one 5 mm tungsten bead using the TissueLyser II system (Qiagen, Hilden, Germany) for 2 min at 25 Hz. RNA extraction was performed with a Maxwell^®^ 16 LEV simplyRNA Tissue Kit (Promega, Madison, WI, USA) in the Maxwell^®^ 16 System (Promega) according to the manufacturer’s protocol and was followed by double strand cDNA (complementary DNA) synthesis applying the SuperScript^TM^ VILO^TM^ Master Mix (Thermo Fischer Scientific, Waltham, MA, USA) for first strand synthesis and the NEBNext^®^ mRNA Second Strand Synthesis Module (New England BioLabs, Ipswich, MA, USA) for second strand.

### 2.3. Sequencing and Sequence Assembly

The cDNA library was prepared with the shotgun methodology using the Nextera XT DNA Library Preparation Kit (Illumina, San Diego, CA, USA) according to the manufacturer’s protocol. Quantification of cDNA was assessed using Qubit 2.0 Fluorometer (Thermo Fischer Scientific) and fragments size range was evaluated using 2100 Bioanalyzer Instrument (Agilent Technologies, Santa Clara, CA, USA). Sequencing was performed on the NextSeq 500 System (Illumina) based on 150 bp paired-end technology [20]. The generated results were assembled de novo by IDBA-UD (k-mers 20, 40, 60, 80 and 100) [21] and SPAdes (k-mers, 21, 33, 55 and 77) [22], and aligned against the non-redundant protein database by DIAMOND [23] with a 10^−3^ e-value threshold. The contigs were inspected at MEGAN6 [24] to identify those corresponding to viral sequences. The IDBA-UD contig was mapped against the raw data using Geneious Mapper at Geneious v.9.1.8 [25]. Then, the analyzed sequence was mapped to reference sequence (NiV; NC_002728) by Geneious Mapper. The raw metagenomic data are available at SRA/NCBI database under the accession number SRR21710685, BioProject PRJNA882858.

### 2.4. Alignment and Identity/Divergence Analysis

The multiple alignment of 86 sequences was performed using Clustal W [26] for both nucleotide (nt) and amino acid (aa) analysis. The PBV partial genome obtained was compared with 72 RefSeq genomes from *Paramyxoviridae* family currently available, nine complete genomes of novel rodent and shrew-borne henipaviruses, and four sequences of Brazilian jeilongviruses. All sequences included in the analysis are available at Appendix A. The partial genome sequence of PBV was deposited in GenBank under the accession number MZ615319.

The identity and divergence matrixes were generated using Geneious v.9.1.8 and MEGA X [27]. The amino acid identity matrix generated by MEGA X using the amino acid substitution p-distances was converted by R programming language in a boxplot graph of intra and intergroups distances/divergence among members of the same genus against each other, and between them and PBV, respectively. Additionally, the corresponding domain and conserved motifs were identified in the amino acid sequence of PBV using InterProScan [28].

### 2.5. Phylogenetic Analysis

The phylogenetic inferences by maximum likelihood (ML) analysis with 1000 bootstrap iterations [29] were built using GTR + F + R6 and LG + F + R10 as substitution models to nucleotide and amino acid, respectively, implemented by IQ-TREE v.2 [30]. The obtained trees were rooted on the midpoint and edited for graphical display using Inkscape v.1.1 [31].

## 3. Results and Discussion

Both assemblers generated similar results. Nine assembled contigs were assigned to *Paramyxoviridae* family by MEGAN6 and four of them were aligned by pairwise identity with NiV reference genome at the L gene portion. Considering its relevance for analysis, the larger contig was selected. The sequence of 2377 nt showed 54.2% pairwise identity with the L gene in the NiV reference genome (12,420 to 14,934 nt), corresponding to 13.02% of the viral genome and 34.14% of the L gene (Figure 2A).

The set of sequences was translated and aligned in order to conduct the phylogenetic inference analysis based on the PBV partial L protein (2244 aa of NiV RefSeq) in the position between sites 393 and 1170 and related to RNA polymerase domain (RdRp; total 777 aa). The alignment is also displayed in Figure 2B with deletion of a major hypervariable region between sites 615 and 701 original residues, and ten conserved motifs among the 15 sequences were highlighted, demonstrating the similarities they share.

The nucleotide and amino acid identities matrixes across the sequences were combined and are showed at Figure 2C. In the alignment matrixes of the partial L genomic sequence, PBV nucleotide and amino acid identities with NiV were 55.29%/55.84% and 55.29%/55.2% for HeV. The low identities in the L gene among distinct henipavirus species and the two original henipaviruses is a common feature, described for the rodent/shrew-borne clade as in MojV [13], Gamak virus and Daeryong virus [15], and observed even in the bat-borne clade as for GhV and *Cedar henipavirus*.

PBV translated amino acid sequence was aligned to a set with all reference genomes of the *Paramyxoviridae* viral family deposited in the RefSeq database. Genomes of novel rodent and shrew-borne henipaviruses [14,15,16], and jeilongviruses recovered from Brazilian bats (Amazon and Atlantic Forest) [32] were also included in the alignment, the latter due to their proximity with *Henipavirus* clade and their Brazilian origin. Both nucleotide and amino acid phylogenetic analysis showed similar topology and the amino acid tree was chosen to represent the phylogeny (Figure 3 and Appendix A). Since a set of 86 sequences was included in the phylogenetic analysis, the *Respirovirus* genus and the *Avulavirinae* and *Rubulavirinae* subfamilies were collapsed in the tree to highlight the results of the main clade, part of the *Orthoparamyxovirinae* subfamily, in which the *Henipavirus* genus is positioned.

The phylogenetic analysis provides evidence of common ancestry shared between PBV and the represented henipaviruses. The sequence clustered as a single branch of a sister clade of the bat-borne and rodent/shrew-borne henipavirus subclades, and clearly separated from the clade of the novel jeilongviruses from Brazil (Figure 3), indicating that PBV is a henipa-like virus. Additionally, the four closest genera to PBV (*Henipavirus*, *Jeilongvirus*, *Narmovirus*, and *Morbillivirus*) were selected to generate a boxplot graph of intra and intergroups divergence (Figure 4), which demonstrates that PBV sequence exceeds the average divergence between members of the *Henipavirus* clade, as well as the other three represented genera, even with the analysis of the conserved RdRp domain. Therefore, along with the resulting topology of the phylogenetic analysis, these data may suggest the existence of a novel group of viruses or even of other lineages of highly divergent henipaviruses, resembling the history of the rodent/shrew-borne subclade, which was initially represented only by MojV [13] and further proven to be more diverse as new members were described [14,15,16,17]. The description of novel divergent henipaviruses could challenge the current taxonomy of the *Henipavirus* clade as a single genus [32], a visible characteristic in the boxplot, in which a high divergence range is observed.

The identification of a genome fragment related to henipavirus sequences is the original genomic evidence of the circulation of henipa-like viruses in Brazil and South America. Previously, just two other sequences of henipa-like viruses were registered in the Americas. The only one available is a 559 bp sequence found in a feces sample of an insectivorous bat of the *Pteronotus parnelli* species collected in Costa Rica in 2010. This sequence corresponds to a fraction of the *L* gene and was classified, by phylogenetic analysis, as a henipa-like sequence of a distinct lineage from henipaviruses circulating in Africa, Southeast Asia, and Australia. The other henipa-like sequence was found in a frugivorous bat of the *Carollia perspicillata* species also collected in Costa Rica, although it is unavailable for analysis. Both bat species are found in the Brazilian Amazon [10], and *Carollia perspicillata* bats are common in the municipality of Peixe-Boi. We did not include the 559 bp (~186 aa) sequence from Costa Rica due to its small length in comparison to PBV sequence.

The two other findings in the Americas are from serological studies in bats. One was performed in a Brazilian tropical savanna biome (Cerrado) in southeastern region of Brazil, in which sera from 76 bats were tested for the presence of cross-reactive NiV antibodies through an in-house developed NiV nucleoprotein ELISA and an IFA on NiV-infected Vero cells. From the 76 samples, 13 were positive, from *Artibeus planirostris* (*n* = 2), *Carollia perspicillata* (*n* = 2), *Artibeus lituratus* (*n* = 1), *Desmodus rotundus (**n* = 1) species, and *Glossophaginae* sp3 (*n* = 6) and *Glossophaginae* (*n* = 1) subspecies [18], all of which are endemic in the Brazilian Amazon. 

The other study was conducted in Trinidad and Tobago and found 28 bat serum samples positive for antibodies reactive with NiV glycoprotein (a type of RBP protein) or fusion protein on ELISA. These bats were of the *Artibeus lituratus*, *Artibeus planirostris trinitatis*, and *Carollia perspicillata* species [19], which were also identified with antibodies reactive to NiV in the Brazilian study [18]. Although none of them provided genomic detection, they support the hypothesis of henipa-like viruses’ circulation in Latin America. 

The detection of PBV is also the original description of henipa-like virus affecting marsupials worldwide. The remaining serological and molecular findings of henipa-like viruses from Latin America were found in bats, similar to most studies conducted in Africa, Southeast Asia, and Australia [10,18,19].

The *Marmosa demerarae* species was previously classified as *Micoureus demerarae*, but after recent taxonomic reorganization based on phylogenetic data, the whole *Micoureus* genus is now considered a subgenus of *Marmosa* [33]. The species is restricted to South America and can be found in French Guyana, Guyana, Venezuela, Peru, and Brazil, including the Brazilian Amazon. These opossums are specialized arboreal and nocturnal with insectivorous and frugivorous diet [34,35], which could occasionally allow the sharing of habitat and food with some bat species, including the species cited in the studies of Trinidad and Tobago and Brazil [18,19]. Therefore, this finding may suggest the possibility of interspecies transmission of PBV to *Marmosa demerarae*, which could be an accidental host. A hypothetical transmission cycle for the reported henipa-like virus is feasible since bats are usually the primary hosts of important henipaviruses, which are the group of viruses most related to PBV.

Another possibility is that PBV is a marsupial-borne henipa-like virus. Recently, the existence of rodent-borne henipavirus was demonstrated in China [13,14], and shrew-borne henipaviruses, from China, South Korea, Belgium, and Guinea [14,15,16,17], which further indicates the possibility that other mammals harbor henipaviruses. Although this matter remains unclear, due to lack of data about PBV genomic organization and ecological features, the marsupial-borne origin hypothesis is still supported by the high divergence of the amino acid sequence and the isolated position of PBV in a single branch of a sister clade to the bat-borne and rodent/shrew-borne subclades in the phylogenetic reconstruction. Further investigations on the diversity of henipa-like viruses among didelphid marsupials might support more robust inferences about this possible new subclade.

The vulnerability of deforested environments such as those at the study site, where wild habitats overlap with rural areas, resembles the conditions in which HeV and NiV outbreaks arose [5]. This emphasizes the importance of further surveillance studies in the research area in an attempt to isolate and characterize the proposed virus and clarify its nature, the relationship with henipaviruses and impact on public health. Furthermore, the role of didelphid marsupials, mainly the *Marmosa demerarae* species, in the hypothetical transmission cycle, must be elucidated. This finding also instigates a wider inclusion of this group of mammals in *Henipavirus* surveillance studies.

## Figures and Tables

**Figure 1 viruses-14-02167-f001:**
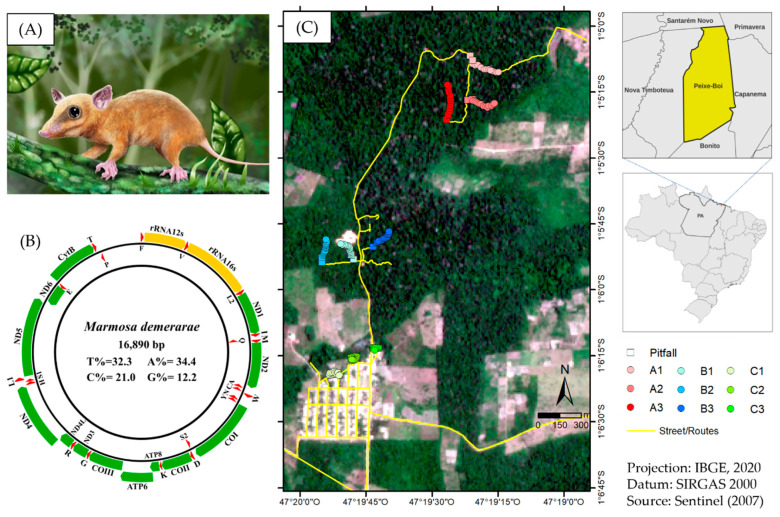
(**A**) *Marmosa demerarae* digital illustration. (**B**) Structural representation of *Marmosa demerarae* mithocondrial DNA. Internal values indicate the content of the nucleotide bases. Yellow, green, and red blocks indicate rRNA, PCGs, and tRNA, respectively. (**C**) Sample collection sites in the proximities of Ananin village, municipality of Peixe-Boi, Pará State, Brazil.

**Figure 2 viruses-14-02167-f002:**
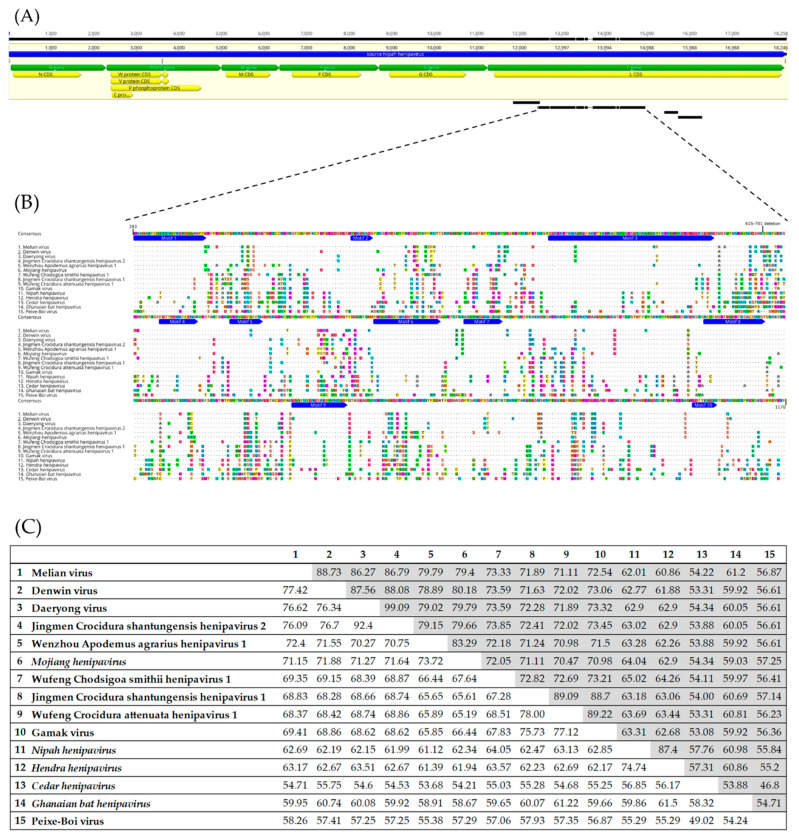
(**A**) Nucleotide alignment of four assembled contigs of PBV with NiV reference genome (NC_002728), and location of each contig within the *L* gene. (**B**) Amino acid alignment across henipaviruses and PBV sequence based on PBV partial L protein position between sites 393 and 1170 (2244 aa of NiV RefSeq). A major hypervariable region between sites 615 and 701 was deleted and ten conserved motifs are highlighted by blue annotations. (**C**) Nucleotide (white) and amino acid (gray) identities among henipaviruses and PBV.

**Figure 3 viruses-14-02167-f003:**
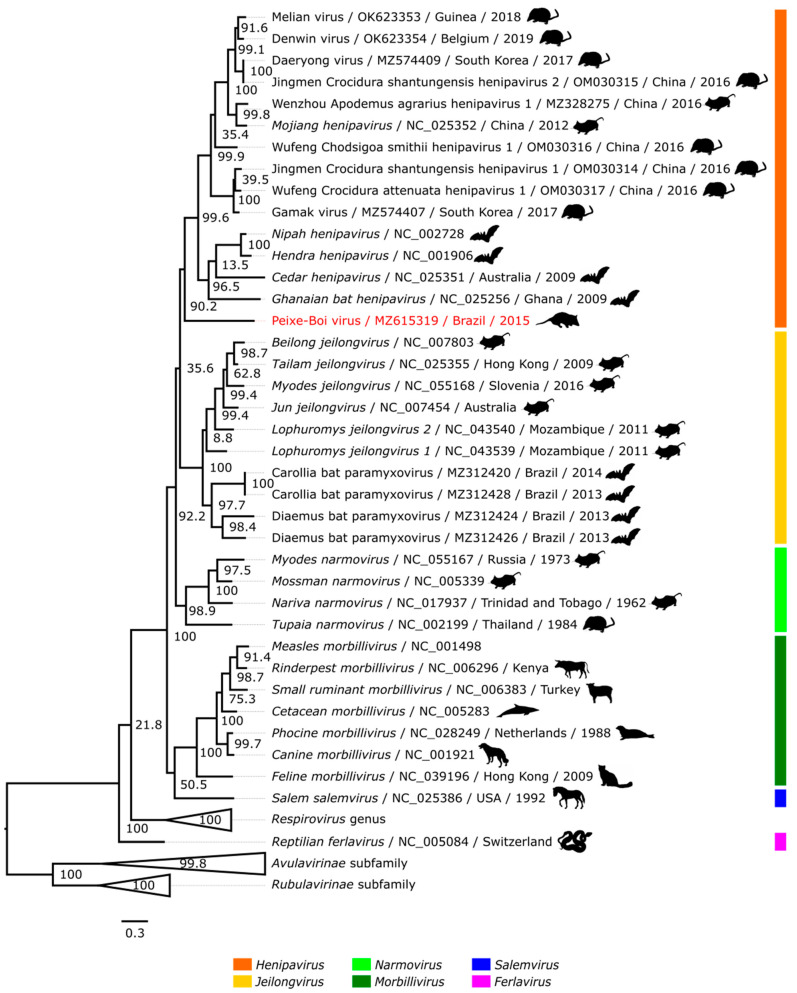
Phylogenetic tree based on the translated *Paramyxoviridae L* gene RefSeq sequences, novel rodent and shrew-borne henipaviruses and Brazilian jeilongviruses sequences. PBV (in red) partial genome was detected in a pooled tissue sample of *Marmosa demerarae*. Each record consists of the virus species/name, accession number, country, and year of detection/isolation and next to the icon of the original host of detection/isolation. *Orthoparamyxovirinae* subfamily is represented with genera indicated by color. The other two subfamilies and *Respirovirus* genus are collapsed.

**Figure 4 viruses-14-02167-f004:**
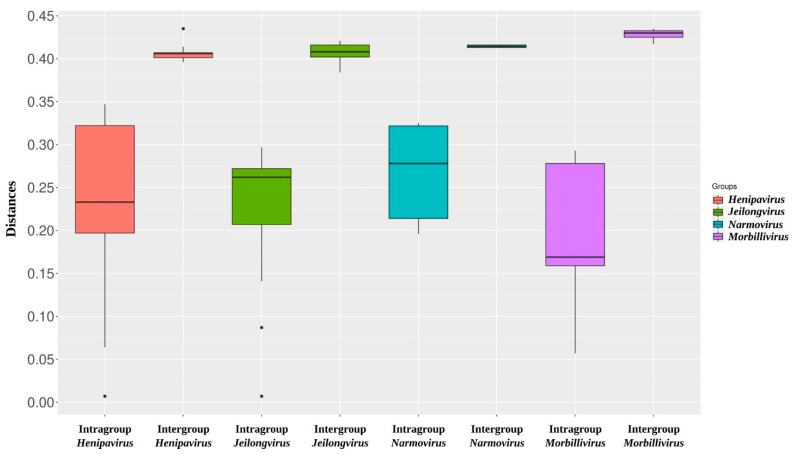
Boxplot graph of amino acid divergence among the members of the four genera closest to PBV (*Henipavirus*, *Jeilongvirus*, *Narmovirus*, and *Morbillivirus*) against each other (intragroup), and between them and PBV (intergroup).

## Data Availability

The consensus sequence is deposited in GenBank under accession number MZ615319.

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
