# Peer review of "First Genomic Evidence of a Henipa-like Virus in Brazil"

_viruses, 2022, doi:10.3390/v14102167_

Round 1
Reviewer 1 Report
This study by Nernandez et al reports a partial genome sequence a novel paramyxovirus from opossums in Brazil. The sequence is genetically related to members of the genus Henipavirus and hence the authors described it as a “henipa-like” virus. The finding could potential be significant, but the current study needs to be further improved to boost the validity and confidence of the findings.
Major issues:
1) The methodology description lacks important detail (see below) and different/better bioinformatic analysis tools should be used
2) Based on the partial sequence presented, it is premature to call it a henipavirus
3) In the phylogenetic analysis, the authors should have included the African and Asian shrew fragments discussed in the manuscript
4) Only nt sequence mapping and comparison was presented in this study, how about amino acid sequence analysis, which is more sensitive in identifying novel viral sequences?
5) No attempt to conduct prevalence study in Marmosa demerarae either by PCR or serology
Minor issues
a) Page 3/line 100-105: no detail given on the timing of sampling and transport. Was any storage buffer used? If so, what buffer?
b) Page 4/lines 122-123: samples were pooled from three animals for NGS analysis. Once the sequence was identified, have the authors gone back to test individual samples by PCR?
c) Page 4/line 133: What primer was used in cDNA synthesis? Random hexmers?!
d) Page 4/138: Since the only evidence in this study is the assembled contig, the choice of de novo assembly really matters. Should consider other well-established tools like SPAdes and Megahit and then make some comparisons.
e) Page 4/line 142: Mapping in Geneious also varies, please specify the mapper (for example, BBMaps or Bowtie2 or Geneious Mapper).
f) Page 4/line 154: 2.377 nt or 2,377 nt?!
g) Page 6/line 193-194: this sentence is hard to understand, please elaborate.
h) Page 7/line 260: Raw NGS data should be deposited
Author Response
Please see the rebuttal letter attached. Our replies to you are the addressed to Reviewer 1.

Reviewer 2 Report
In this manuscript, Hernandez et al. have described genomic evidence of a henipa-like virus from opossums which they have named after the Peixe-Boi municipality from which they captured the animals.
While the finding itself is certainly interesting, especially in light of the recent documentation of Langya henipavirus likely being the cause of human disease in China, the reviewer has several questions, concerns, and areas for revision regarding the methodology and presentation of this study.
1. Lines 100-113. The three opposums captured in this study are presumably part of a larger animal surveillance study- what other animals species were captured, and were samples from such studies blindly subjected to next-generation sequencing in search of henipa-like viruses as well?
2. Line 106. The digital illustration of the Marmosa demararae opossum in Figure 1 is insufficient and probably altogether unnecessary. If the authors would like to include an image, it should be that of the actual animal.
2. Line 122. The fact that tissue samples from the 3 animals were pooled prior to nucleic acid extraction and downstream cDNA construction/library prep and NGS processes is problematic. This brings up the question as to whether the viral sequence reconstructed from NGS is potentially an amalgamation of different viral sequences instead of representing one true henipa-like virus sequence.
3. Lines 156-158. It is intriguing that the authors purportedly could not incorporate Cedar henipavirus into their phylogenetic analysis. Given the major divergence of sequences used across the Paramyxoviridae family, it seems strange that incorporating another bona fide henipavirus sequence into the analysis would confound the alignments.
4. Lines 179-181. The authors need to provide a table documenting both nucleotide and amino acid identity across henipaviruses in comparison to their single 2,377 nt reconstructed sequence from the opossum.
5. Lines 185-193. Prior to (or even following) the NGS analysis, did the authors conduct conventional consensus PCR of conserved pan-paramyxovirus and pan-respiro-morbilli-henipavirus of the pooled samples (as per Tong et al., 2008 https://journals.asm.org/doi/10.1128/JCM.00192-08)? Conducting these consensus PCRs to compare with the many paramyxovirus sequences known at both the nucleotide and amino acid levels are absolutely critical to include in such a study. This should be done independently of their phylogeneic analysis of the full 2,377 nt sequence analysis (which also should include amino acid identity comparisons).
6. Line 203. In reference to the de Araujo manuscript, the authors should particularly mention the Glossophaginae subspecies (sp.3) in which a majority of Nipah virus N protein ELISA positive sera were identified.
There are serious concerns about omissions in this manuscript as it currently stands- the authors should comprehensively address the points mentioned in the above in a major revision in order to be considered for publication.
Author Response
Please see the rebuttal letter attached. Our replies to you are the addressed to Reviewer 2.

Round 2
Reviewer 2 Report
The authors have made some significant improvements to the manuscript. The reviewer has included a pdf of minor grammatical edits to improve the flow of the manuscript.
The reviewer has a few comments below:
1) Please include the nucleotide phylogenetic tree in the Supplementary Materials so that readers can confirm the similarity with that of the amino acid-based phylogenetic inference in Figure 3.
2) Lines 91-92: Please cite the two serological studies in the Americas in the introduction.
2) Lines 237-266: The order of citation of the two serological studies in the Americas should be reversed. The de Araujo et al., manuscript should be described first, as it was completed 3 years prior to the study in Trinidad and Tobago. It is more appropriate to state that the bat species identified from the Trinidad/Tobago study confirmed some of those already identified by the Brazilian study 3 years earlier. Please change the wording to factually reflect the order of studies.

Author Response
Dear Reviewer,
Thank you for your suggestions. We made the corrections and added the nucleotide phylogenetic tree in the supplementary material. The updated version of the manuscript is available.
Just to be clear, we made a simple adjustment in the figure 3 using Inkscape just to move some hosts icons that were very close to their branches descriptions. No philogenetical changes were made.
Kind Regards,
Ana Cecília Ribeiro Cruz